# Progress of Standardization of Urban Infrastructure in Smart City

**Jin Wang [1], Chang Liu [1], Liang Zhou [1], Jiangpei Xu [1], Jie Wang [1] and Ziqin Sang [2],***

[1] State Grid Hubei Electric Power Research Institute, Wuhan 430070, China
[2] China Information Communication Technologies Group, Wuhan 430074, China
* Correspondence: zqsang@wri.com.cn; Tel.: +86-27-87694040

**Abstract:** After the Smart City initiative was put forward, cities all over the world started the pilot practice of developing Smart Cities. This triggered a series of thoughts: what is a Smart City, how do we determine the scope of work of a Smart City, and how do we formulate a new strategic agenda of the Smart City to make city smarter and more sustainable? The answer lies not only in finding Smart City solutions, but also leads to the research on the definition of Smart City terminology and the determination of corresponding tasks. Stakeholders of Smart City (e.g., policy makers, municipalities, solution providers, industry, and academia) develop technical and management standards for these tasks jointly. This paper reports the standardization planning on Smart City by the international standardization development organizations (SDOs), that is, the standardization framework of Smart City. It also presents one of the important aspects, namely, the progress of standardization activities on urban infrastructure that are being carried out by the International Telecommunication Union (ITU) via its Study Group 20, in supporting the adoption of information and communication technologies (ICTs) in Smart City. These standards include the classification of urban infrastructure, the interoperability between urban infrastructure and smart city platforms, and the requirements of detailed infrastructure from the perspective of ICT and the Internet of things (IoT). This paper also provides the use cases of application of some standards in global cities.

**Keywords:** information and communication technologies; internet of things; smart city; smart city platform; standardization; urban infrastructure

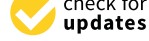



## 1. Introduction

After IBM proposed a smarter planet in 2008 [1], many countries began to introduce the Smart City concept in urban construction. The Ministry of Science and Technology of China launched a Hi-Tech Research Programme project (the 863 Programme) in 2010 to start Smart City research and practice. As one of the two pilot cities of the project, the city of Wuhan launched a bidding procurement of the Wuhan Smart City conceptual design project in March 2011 [2] and a bidding procurement of the Wuhan Smart City master planning and design project in May 2011 [3], which might be the first Smart City commissioned research projects anywhere in the world. The purpose was to determine the concept of Smart City, and to clarify its scope of work and construction tasks. In particular, it was pointed out in the biddings that the design schemes were the basis and guiding documents for the subsequent construction and implementation of the Smart City pilot project. There were similar studies in Europe [4]. The concept of Smart City was introduced as a strategic device to encompass modern urban production factors within a common framework. Since then, more and more cities have begun master planning, design and pilot projects of Smart Cities [5]. The cities unanimously stressed the urgent need for Smart City standards. The rationale is as follows: without standards, the objectives of Smart City construction cannot be determined. If there are no unified standards, repeated construction and a waste of resources may not be avoided. Moreover, due to the lack of standards, it is difficult to evaluate the achievements of Smart City construction.

This urgent demand for standards has been partially responded to by some standards organizations, forums, and consortia. For example, the China Communications Standards Association (CCSA) started the exploration of standardization on Smart City and carried out some standard work items in 2012 [6]. ITU organized a forum on "Greener Smarter Better Cities" during "Green Standards Week" in September 2012 [7], and released a document "Call to Action on Smart Sustainable Cities". It invited stakeholders to create a focus group to analyse smart solutions that may be standardized, and to identify best practices that can facilitate the implementation of such solutions in cities. This has gradually opened up the standardization activities of Smart Cities.

In this paper, we summarize the international standardization activities on Smart City in Section 2, present the progress of standardization on urban infrastructure in Section 3, and provide use cases of application of some standards in global cities in Section 4.

## 2. International Standardization Activities on Smart City

### 2.1. Pre-Research on International Smart City Standardization

As a member of ITU Telecommunication Standardization Sector (ITU-T), we jointly proposed to establish the Focus Group on Smart Sustainable Cities (FG-SSC) in February 2013. The International Electrotechnical Commission (IEC) established the Smart City System Evaluation Group (IEC/SEG 1) in June 2013 and launched the Smart City strategy study in its Market Committee. The Information Technology Standardization Joint Technical Committee (ISO/IEC JTC 1) established the Smart City group at its plenary meeting in November 2013. The International Organization for Standardization (ISO) established the Advisory Group (AG) on Smart Cities at the plenary meeting of the Technology Management Bureau (TMB) in February 2014. The ISO Technical Committee for Sustainable Development of Communities/Sub Technical Committee for Infrastructure of Smart Communities (ISO/TC 268/SC 1) also carried out pre-research on the standardization of smart communities in the same period.

Usually, different standardization development organizations (SDOs) carry out standardization work on a hot topic in parallel. Given their expertise, they focus on different aspects of the standardization of Smart Cities. However, some duplication and overlap cannot be avoided. Through the establishment of an ISO-IEC-ITU joint Smart City task force (J-SCTF) [8], we coordinate our work in order to minimize duplication, as well as to propose the development of a common text.

### 2.2. Standardization Roadmap

Through the FG-SSC, ITU brought together experts from among its membership, including policymakers, academia, technical experts, representatives from the private sectors and other key stakeholders to formulate guidelines of constructing Smart Cities, as well as the needs of standardization [9].

The focus group put forward a series of work objectives, and also produced a series of achievements, in particular confirming the core role of ICT in Smart City construction [10]. Another outstanding achievement of the focus group was the definition of the term "Smart Sustainable Cities" (SSC) [11,12], which was later included in the vocabulary database of the ITU-T Standardization Committee for Vocabulary [13]. This definition has also been adopted by more than a dozen United Nations agencies, which coordinate the initiative of "United for Smart Sustainable Cities" (U4SSC) to promote global Smart Cities in achieving the UN Sustainable Development Goals [14].

We proposed a framework of standards for SSC (see Figure 1) in FG-SSC deliverables [15,16]. The SSC standards can be classified into four categories: (i) SSC management and assessment; (ii) SSC services; (iii) ICT; and (iv) buildings and physical infrastructure. We also proposed a standardization roadmap, taking into consideration the activities undertaken by the various SDOs, forums and consortia. The relevant standards and documents have also been collected and listed [17].

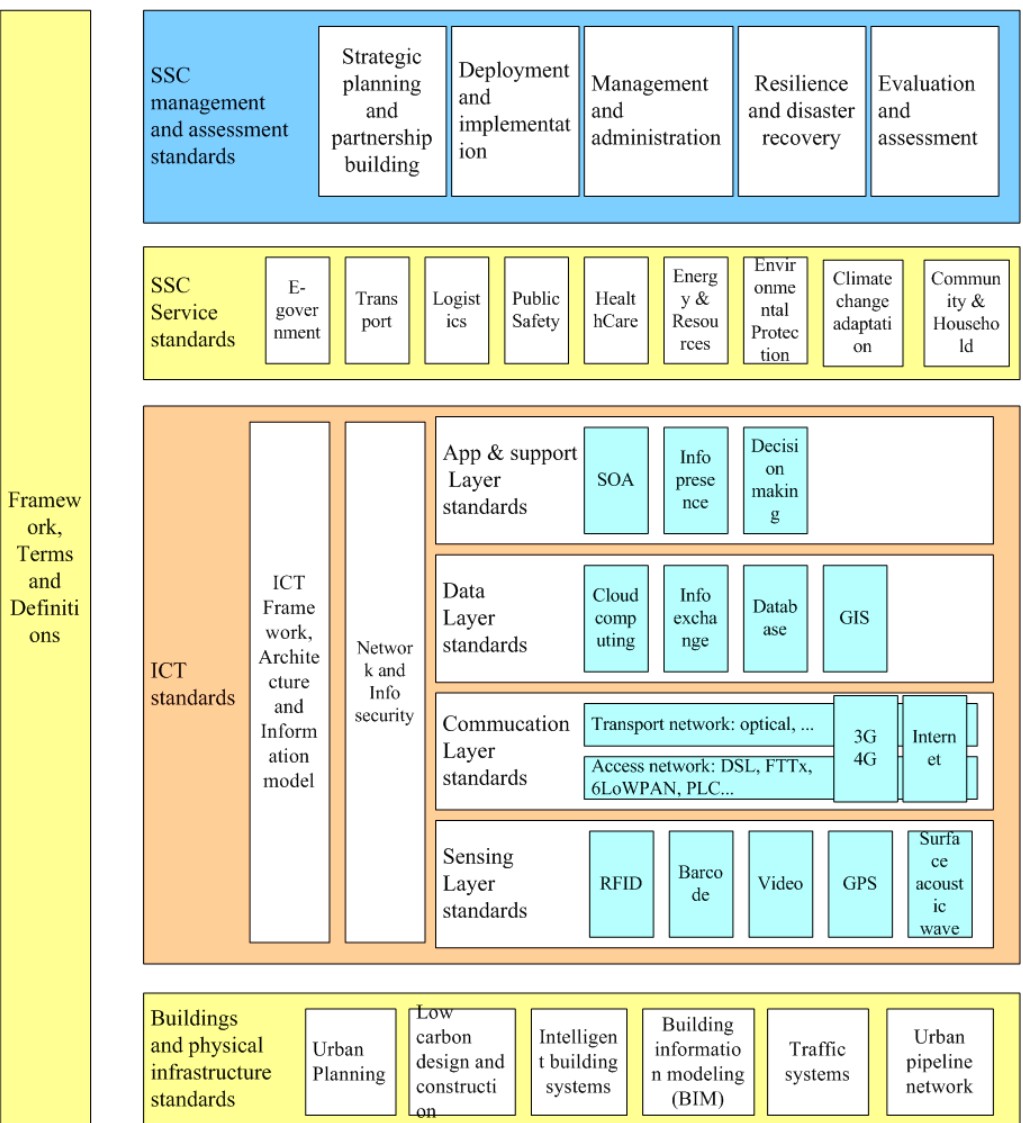

**Figure 1.** Framework of SSC standards.

The work of FG-SSC has concluded in May 2015 with 21 deliverables of Technical Reports and Technical Specifications on overview of SSC, SSC definition, city leaders' guide, master plan, stakeholder analysis, infrastructure, ICT architecture, cybersecurity, open data, smart building, smart water management, climate change adaptation, electromagnetic field consideration, integrated management, key performance indicators, and other matters [18].

### 2.3. ITU-T Study Group 20

The work of a focus group is usually continued by one or more study groups in the ITU-T. Established in 2015, the ITU-T Study Group 20, titled "Internet of Things and Smart Cities and Communities" (ITU-T SG20), is the leading study group on IoT and its applications, as well as Smart Cities and Communities. Currently, SG20 has seven Questions under study [19].

Table 1 lists ITU-T SG20's key standardization areas in IoT and Smart Sustainable Cities [16,19]. Each category contains multiple international standards developed by SG20 that support the deployment of ICT in SSC.

**Table 1.** ITU-T SG20 key standardization areas in IoT and Smart Sustainable Cities.

| ITU-T Y.4000 Series | Category |
|---|---|
| Y.4000–Y.4049 | General |
| Y.4050–Y.4099 | Definitions and terminologies |
| Y.4100–Y.4249 | Requirements and use cases |
| Y.4250–Y.4399 | Infrastructure, connectivity and networks |
| Y.4400–Y.4549 | Frameworks, architectures and protocols |
| Y.4550–Y.4699 | Services, applications, computation and data processing |
| Y.4700–Y.4799 | Management, control and performance |
| Y.4800–Y.4899 | Identification and security |
| Y.4900–Y.4999 | Evaluation and assessment |

The flagship seminar of SG20, "Digital transformation of cities and communities", is a series of webinars organized by ITU, together with other organizations and United Nations agencies, running from September to December 2021 [20]. The purpose is to discuss topics related to the digital transformation for cities and communities and their standardization, and to investigate the expanding role of digital transformation in driving innovation, sustainable growth and inclusion, as well as to respond to crises in cities and communities.

*2.4. Interoperability of Smart City Platforms and Urban Infrastructure*

When SG20 was established, the new work items on Smart City standards were created in accordance to the standardization framework illustrated in Figure 1. Considering the importance of core ICT infrastructure in urban information construction, our team proposed a work item on Smart City ICT frameworks in October 2015, whereas the team from Spain proposed another work item on ICT systems for Smart City management in January 2016. After several meetings, SG20 created two work items on draft Recommendations on Smart City platforms (SCPs) and their interoperability. ITU-T published these two Recommendations on SCPs in February 2018 [21,22].

The urban information systems collect data about the status of the city. These systems are managed by different types of control systems (such as IoT platform, Supervisory Control and Data Acquisition (SCADA) system, Big Data platform, etc.). In most cases, these vertical systems are separated, non-standardized and closed, and it is difficult to share resources and data. By introducing a common Smart City platform, multiple vertical systems are integrated and optimized to provide interaction between urban information systems to support various functionalities of urban services and ensure the convenience, security and scalability of the platform [22].

An SCP integrates city platforms and systems (i.e., SCP functions) directly, or through open interfaces between SCP and external providers, to offer the urban operation and services supporting the functioning of the city services. These interfaces are shown in Figure 2 as red arrows, as detailed in the following list:

- Services interface, which connects the SCP's Services Support Functions and the external services and applications providers.
- Interoperability interface, which connects the SCP's Data/Knowledge Functions interoperability and the external database and computation systems.
- Acquisition interface—which connects the SCP's Acquisition/Interconnection Functions and the external sensing and infrastructure systems.

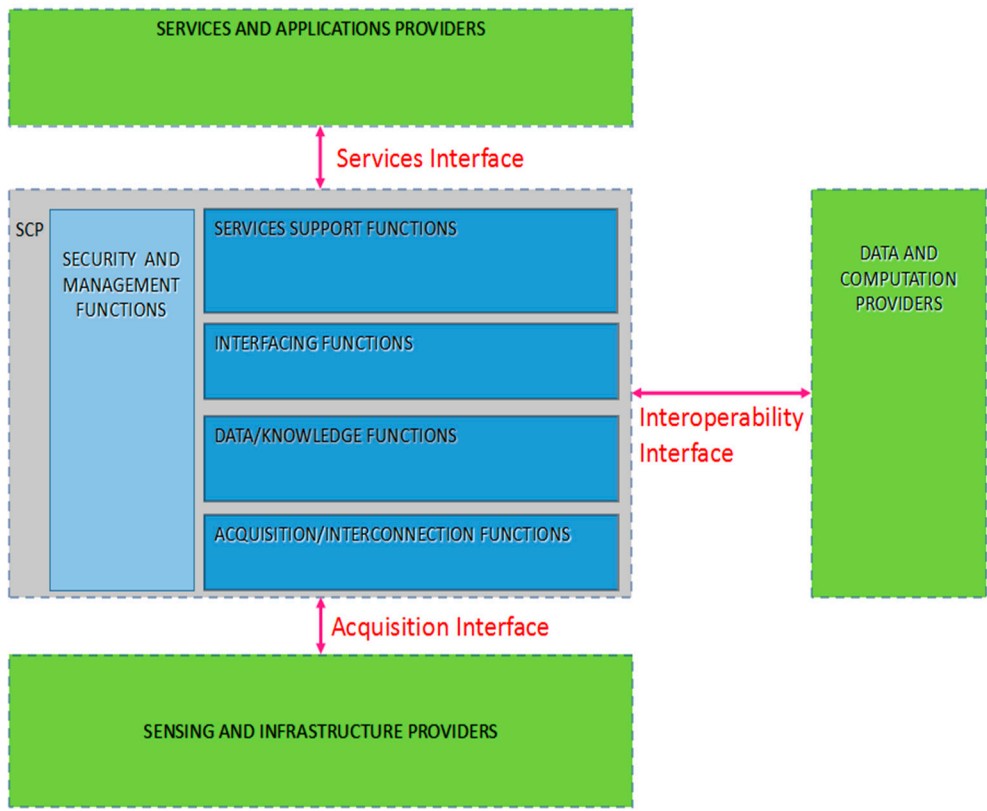

**Figure 2.** Reference framework of SCP.

## 3. Standards on Urban Infrastructure

When the Smart City goals were proposed, it was not clear how to improve the intelligent management through emerging technologies, or which urban infrastructure could be improved. We collected information on existing city infrastructure and released a technical report as a deliverable of FG-SSC [23], in order to figure out which infrastructure could be improved and which technology could be used. When SG20 develops standards based on relevant technical reports, we still need to answer a basic question: what is the definition of Smart City infrastructure? That is, what is the urban infrastructure within the scope of Smart City, and what is not? We then proposed a standard work item in October 2015—the overview of Smart City infrastructure [24]. This work item has lasted for a long time, and the content has been gradually enriched. Many work items have been derived. Consequently, urban infrastructure standardization has been expanded into a series.

Generally speaking, the urban infrastructure of Smart City should comprise man-made facilities. Naturally formed environments such as mountains, rivers, grasslands, forests, and lands are not urban infrastructure. The urban infrastructure of Smart City is identified as a variety of systems such as energy systems, water systems, transportation systems, communication systems, disaster risk-reduction systems, healthcare systems, and information transmission systems. Smart Cities need to use "intelligent" technologies to provide each physical infrastructure with "intelligence", so as to improve the intelligence level of the overall urban infrastructure. At the city level, it needs to implement appropriate systems and measures in order to meet the United Nations' Sustainable Development Goals [24,25].

Based on the architecture in Figure 2, ITU-T SG20 has issued, or is developing, a series of Recommendations, including definition and classification of urban infrastructure, management of urban assets, sensing and data collection system for city infrastructure, event monitoring for city infrastructure, and some specific application systems, such as civil engineering health management systems, construction site management systems,

electricity infrastructure monitoring systems, fire water supply monitoring systems, and smoke detection monitoring systems.

### 3.1. Classification of Urban Infrastructure

In ITU, city infrastructure is defined as "the interconnected structures that enable people to get the resources they need in the city, and also the interconnected structures to provide public services for social and economic activities in the city" [24]. The basic infrastructures for building and operating SSC are categorized as:

- energy infrastructure;
- water supply and drainage infrastructure;
- transportation infrastructure;
- post and communication infrastructure;
- disaster risk-reduction infrastructure;
- cultural, sports and educational infrastructure;
- healthcare infrastructure; and
- social welfare infrastructure.

At city level, the infrastructure should be managed through ICT, and the sensing data can be collected by sensors deployed within these infrastructures.

### 3.2. Management of City Assets

From the perspective of assets, city infrastructure is also considered as a city asset [26]. The typical kinds of physical city assets may include, but are not limited to environmental sanitation facilities, water supply and drainage facilities, energy facilities, transportation facilities, postal and telecommunication facilities, and disaster risk-reduction facilities (shown in Figure 3).

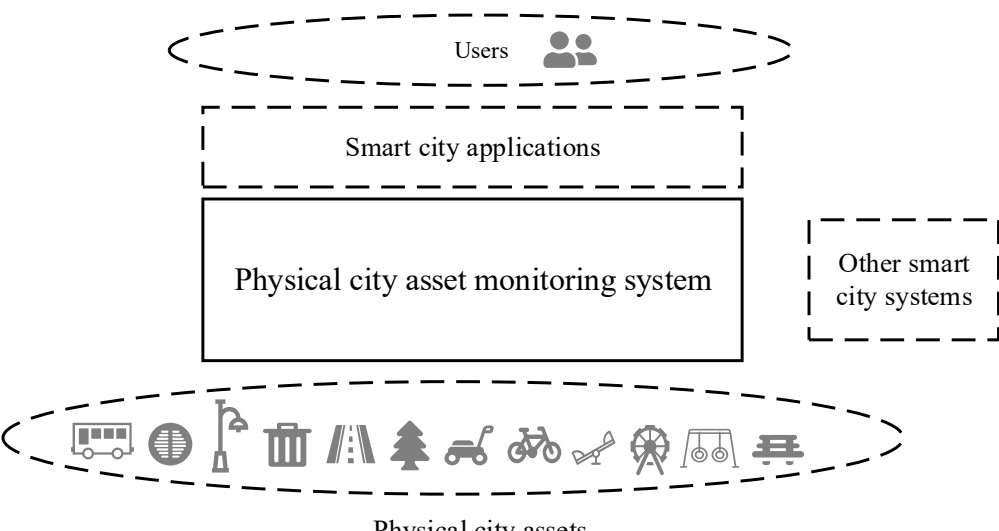

**Figure 3.** Overview diagram for physical city asset monitoring system.

Physical city assets keep evolving with technology and city development. With the support of IoT, the safeguarding, maintenance, and management of city assets can enable add a new service experience to the diversity of users, including, but not limited to, citizen and city asset operators.

The capability framework of city asset monitoring system is composed of city asset monitoring device capabilities, IoT device management capabilities, city asset monitoring service capabilities, network connectivity management capabilities, and identification capabilities.

### 3.3. Sensing and Data Collection System for City Infrastructure

Shown in Figure 4, sensing and infrastructure system consists of a sensing and data collection system (SDCS) and city infrastructures. An SDCS monitors the status and collects information of different kinds of city infrastructures, controls and manages the sensing devices attached to those city infrastructures, and provides corresponding information to SCP [24].

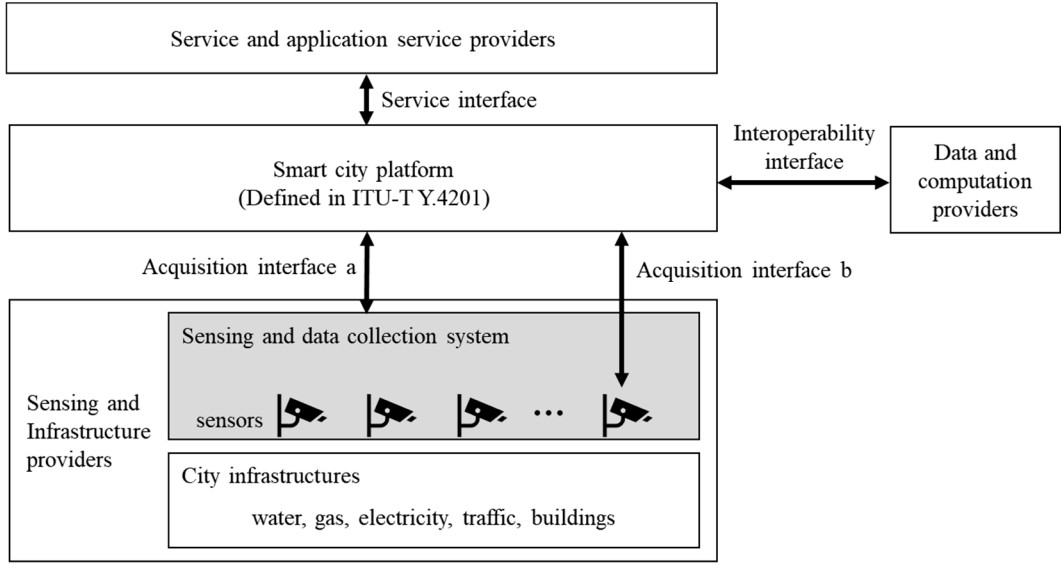

**Figure 4.** Interoperability of SCP and city infrastructure.

The SDCS can both provide the SCP with raw sensing data and can work as an agent to control and acquire data from a sensing device to the SCP. When the SCP acquires data directly from different kinds of sensing devices, the SDCS is working on transparent transmission mode and the sensing devices transmit data through the acquisition interface b to provide the SCP with raw sensing data. When there is a large deployment of sensing devices among city infrastructures, the SCP is unable to manage all the sensing devices. In such a case, the SDCS needs to be used as an agent to control and acquire data for the SCP, and it works on agent transmission mode and transmits processed sensing data via acquisition interface a.

### 3.4. Event Monitoring for City Infrastructure

Considering the needs of event monitoring of disasters, a disaster monitoring system (DMS) is defined and the DMS data are extracted from the sensing and data collection system (shown in Figure 5) [27]. The DMS aims to provide high-precision, multitemporal, and full-range sensing of disaster events. The structure and elements of sensing capabilities are described by a disaster monitoring metamodel [28].

This standard meets the two needs of the above SDCS when encountering specific disaster events: (1) to effectively discover the suitable sensors for specific disaster event from a large number of sensors; and (2) to collaborate with multiple sensors for disaster event monitoring.

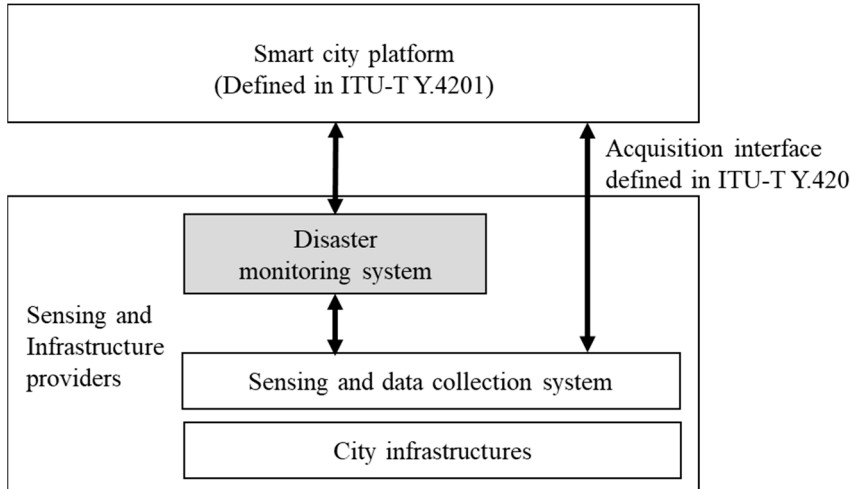

**Figure 5.** Interoperability of SCP, DMS, and city infrastructure.

*3.5. Industry-Specific Infrastructure Monitoring*

In addition to the general standards of urban infrastructure, SG20 has also formulated, or is developing, some standards for industry-specific infrastructure monitoring such as civil engineering infrastructure health monitoring, construction site management, electricity infrastructure monitoring, fire water supply monitoring, and smoke detection monitoring. These industry-specific standards usually have a common architecture, but they also have their own characteristics. By formulating and publishing these standards, the construction and maintenance of Smart Cities can be guided better.

3.5.1. Civil Engineering Infrastructure Health Monitoring

The civil engineering infrastructure refers to the large-scale structures that make up roads, bridges, and tunnels. The maintenance of civil engineering infrastructures is mainly based on periodic visual inspections and diagnoses by specialists to maintain the safety and integrity of the infrastructure. These specialists need a high degree of experience and technical know-how, as maintenance requires relevant human resource allocation and specific education and qualifications. Utilization of objective data collected from the civil engineering infrastructure using IoT capabilities leads to advanced and more efficient maintenance, while enhancing and rationalizing the human inspections and diagnostic work [29].

This Recommendation addresses IoT-based civil engineering infrastructure health-monitoring systems specifying the following:

- a reference model of IoT-based civil engineering infrastructure health monitoring systems; and
- requirements specific to IoT-based civil engineering infrastructure health-monitoring systems.

3.5.2. Smart Construction Site Management

Different from the monitoring of civil engineering infrastructure, smart construction site management focuses on the construction site scenario. It collects the information of construction processes through a smart construction site device (SCS-D) and transmits it to the smart construction site service platform (SCS-SP), so as to improve the construction process management, safety monitoring, and intelligent production [30].

SCS services are introduced to support users by achieving the following goals:

- to help realize the visualized and intelligent management of the construction project, by carrying out construction process trend predictions and emergency plans;
- to improve the level of information-based management of the construction project;

- to realize green construction and ecological construction;
- to potentially improve the modes of interaction, working, and management of any parties involved in the construction project;
- to eliminate potential waste of resources, and continuously improve the quality, scheduling, and cost of the project, with minimal resource input; and
- to maximize the benefit of the project, so as to meet the needs of customers and to ensure a maximized value.

### 3.5.3. Electric Power Infrastructure Monitoring

In electric power systems, electric power infrastructure (EPI) refers to the infrastructure in four processes, namely generation, conversion, transmission, and distribution of electricity. EPIs are complex and numerous in each process which requires different professional enterprises to carry out daily maintenance. The traditional daily maintenance of EPIs is based on regular inspection and diagnosis by specialists [31].

The objective of EPI health monitoring is to measure the health status of EPIs. By obtaining and analysing data, specialists can understand the change and development process of health status indicators over time. These health status quantities usually include temperature, humidity, deformation, insulation, fire protection, security, among others. An IoT-based electric power infrastructure monitoring system (IoT-EPIMS) is an advanced and efficient means to obtain the information and data required for the health maintenance of EPIs and an auxiliary means for the management of EPIs. Consequently, it can maintain the safe and stable operation of electric power systems and improve the comprehensive management level of power systems.

The scope of this Recommendation includes

- a reference model for IoT-based electric power infrastructure monitoring system;
- requirements specific to IoT-based electric power infrastructure-monitoring systems; and
- use cases of IoT-based electric power infrastructure-monitoring systems are provided in Abbreviations and Acronyms.

### 3.5.4. Monitoring of Water System for Smart Fire Protection

The water supply system is one of the most important parts of firefighting systems. The smart fire protection facilities based on IoT technologies are developed to monitor the water supply status in real time. It is required to accurately obtain the key data of a natural water source, a manmade water source, and a municipal fire hydrant. It will realize the digital management of fire extinguishing water source, providing reliable information for firefighting and rescue operations, enhance the availability of fire control information, and enable the on-site fire brigade to quickly develop targeted water supply plans [32].

This Recommendation introduces smart fire protection (SFP) and presents the monitoring of water systems (MWS) for SFP. This recommendation specifies requirements, functional architecture and functional components of MWS for SFP.

### 3.5.5. Smart Fire Smoke Detection

The fire smoke detection service is usually deployed in indoor environments such as residential buildings, factories, shopping malls, hotels, and office buildings. With the development of society and the economy, the fire smoke detection service is playing a more and more important role in people's lives [33]. The smart fire smoke detection (SFSD) service not only detects smoke concentrations through sensors, and triggers a fire alarm when it reaches a certain threshold to prevent disaster, but also uses the network to send the alarm information to the cloud platform, thus notifying relevant departments and personnel in a timely manner through web/APP/SMS/voice/instant message client, and so on. The SFSD service can provide many benefits, including efficient maintenance and management, real-time alarm reports, real-time fault reports, and good service experience.

The scope of this Recommendation includes

- introduction to the SFSD service, including the issues of the traditional fire smoke detection service and the benefits of the SFSD service;
- requirements of the SFSD service for SFSD device capabilities and SFSD platform capabilities;
- functional architecture of SFSD devices and the SFSD platform within the SFSD service; and
- implementation and deployment model.

## 4. Use Cases of Standard Application

### 4.1. Use Cases of ITU-T Y.4900 Series

When developing, and after releasing, a standard, it is necessary to check whether it is applicable in practice. U4SSC can be regarded partly as an experimental and verification organization of ITU-T Smart City standards. By using ITU-T Y.4900 series standards [12,34–36], more than 150 cities around the world have evaluated and verified their achievements in Smart City construction, and provided U4SSC with verification reports [37].

It should be noted that Dubai, in cooperation with ITU-T FG-SSC in 2015, was the first city in the world to use FG-SSC Technical Specifications on KPIs [18] for SSC evaluation. In 2016, Singapore became the second to participate in the cooperation. Both cities have reported their practice on the SSC evaluation, especially the applicability of some indicators, which have enabled us to revise the series FG-SSC Technical Specifications on KPIs, and to make it a series of ITU-T Y.4900 Recommendations in 2016. Both Dubai and Singapore case studies can be downloaded from the ITU website [38].

### 4.2. Use Cases of ITU-T Y.4201 and Y.4200

ITU-T Y.4201 provides a set of high-level requirements that define the functionalities and interfaces of an SCP while ITU-T Y.4200 specifies the interoperability requirements of each function and interface listed in Y.4201 [21,22]. Both Recommendations advance sustainability and resilience in Smart Cities by providing the blueprint of an open and interoperable SCP that is capable of addressing a wide range of city challenges including but not limited to urban sensing, infrastructure management, climate change, and citizen-centered integrated services. Such a platform is the digital foundation for circulating data collected by different sensor networks and other sources and translating them into actionable insights that support city stakeholders in making better decisions.

In 2015, the provincial government of Hubei in China developed a Sharing Application Platform for Government Information Systems [39]. This information-sharing application platform was developed in accordance with the framework and specifications proposed in Y.4201 and Y.4200. The purpose of this platform is to coordinate captured data and apply them in order to improve city services. Through the Sharing Application Platforms of Government Information Systems, many of the isolated data silos have been eliminated. The work efficiency of the digital applications has been improved greatly. Since its construction in 2015, the platform has been instrumental in realizing the "digital government" vision of the Hubei Provincial Government. The platform is now actively involved in improving medical health, maximizing the efficiency of public transportation, monitoring upcoming extreme weather events, improving disaster management, and reinventing the education system along with other public services. At present, there are more than 700 digital government applications from more than 70 government departments running on this platform.

Benidorm in Spain is a popular tourist destination located close to the Mediterranean Sea. The city has a population of more than 75,000 and welcomes a huge number of vibrant visitors all year around. To enhance tourists' experience in the city, the government has implemented an intelligent management system, a Smart Destination Platform [40], for tourism that aims at using digital technologies to deliver tourism information in near real time. This Smart Destination Platform is constructed based on the specifications listed in Y.4201 and Y.4200. The smart platform is an open SCP that is able to receive

information from a wide range of external communication sources including TripAdvisor, Twitter, and Airbnb. Collected data are then analyzed and redistributed to different city departments that would like to see an improvement in their water cycle, energy efficiency, traffic control, and other aspects of the city. The Smart Destination Platform has made remarkable improvements to the tourism industry in Benidorm. For example, the number of tourists attending sporting events has increased dramatically. Thanks to the information that is made available to them by analyzing real time data through the Smart Destination Platform, tourists are able to choose to attend their favorite sporting events, from running and cycling to soccer matches. The data gathered by the Smart Destination Platform also indicate that there is a strong demand for environmentally friendly commuting options in the city. At present, there are more than 24 other tourist destinations that have implemented this type of Smart City platform in Spain, thus fulfilling the country's vision of a digital future and its digital agenda.

In 2019, ITU applied the 2019 GEC Catalyst Awards of the Global Electronics Council based to these two standards and their applications in China, Spain and several other countries [41].

## 5. Conclusions

Rapid urbanization is the trend in most developing countries. At present, the investment in new infrastructure is far from over, and the developed countries are also facing challenges in the maintenance and upgrading of infrastructure. With the application of artificial intelligence and other technologies, Smart Cities have more available technologies. The goal of Smart City is not static. It will grow with the needs and development of economy and society, that is, the so-called "spiral rise".

So far, some framework standards of urban infrastructure have been published with the efforts of standardization carried out by the international SDOs, as well as the cooperation with some national standardization bodies and industrial standardization organizations. We expect an increasing number of stakeholders to participate in more standardization activities. For the prospect of infrastructure standardization, we look forward to the emergence of more and more detailed standards (such as data modeling and infrastructure anomaly detection) and urban infrastructure evaluation standards, so as to better implement the goal of "making cities and communities better".

**Author Contributions:** Writing—original draft preparation, J.W. (Jin Wang) and C.L.; investigation, L.Z. and J.X.; data curation, J.W. (Jie Wang); writing—review and editing, Z.S. All authors have read and agreed to the published version of the manuscript.

**Funding:** The research of the corresponding author was funded by the Ministry of Science and Technology of China (the 863 Programme 2018YFB2100500).

**Institutional Review Board Statement:** Not applicable.

**Informed Consent Statement:** Not applicable.

**Data Availability Statement:** Not applicable.

**Acknowledgments:** The work of the corresponding author is supported by the Hi-Tech Research Programme project (the 863 Programme 2018YFB2100500).

**Conflicts of Interest:** The funders had no role in the design of the study; in the collection, analyses, or interpretation of data; in the writing of the manuscript, or in the decision to publish the results.

## Abbreviations and Acronyms

This paper uses the following abbreviations and acronyms:

| | |
|---|---|
| CCSA | China Communications Standards Association |
| DMS | Disaster Monitoring System |
| EPI | Electric Power Infrastructure |
| FG-SSC | Focus Group on Smart Sustainable Cities |
| IBM | International Business Machines Corporation |
| ICT | Information and Communication Technology |
| IEC | International Electrotechnical Commission |
| IEC/SEG 1 | IEC Smart City System Evaluation Group |
| IoT | Internet of Things |
| IoT-EPIMS | IoT-based Electric Power Infrastructure Monitoring System |
| ISO | International Organization for Standardization |
| ISO/IEC JTC 1 | ISO/IEC Information Technology Standardization Joint Technical Committee |
| ITU | International Telecommunication Union |
| ITU-T | ITU Telecommunication Standardization Sector |
| J-SCTF | ISO-IEC-ITU Joint Smart City Task Force |
| MWS | Monitoring of Water System |
| SCADA | Supervisory Control and Data Acquisition |
| SCP | Smart City Platform |
| SCS-D | Smart Construction Site Device |
| SCS-SP | Smart Construction Site Service Platform |
| SDCS | Sensing and Data Collection Systems |
| SDG | Sustainable Development Goal |
| SDOs | Standardization Development Organizations |
| SFP | Smart Fire Protection |
| SFSD | Smart Fire Smoke Detection |
| SG20 | ITU-T Study Group 20 |
| SSC | Smart Sustainable Cities |
| TMB | Technology Management Bureau |
| U4SSC | United for Smart Sustainable Cities |
| UN | United Nations |

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
