# Peer review of "Progress of Standardization of Urban Infrastructure in Smart City"

_standards, doi:10.3390/standards2030028_

Round 1

Reviewer 1 Report

Comments and Suggestions for Authors

Review: Progress of Standardization of Urban Infrastructure in Smart City

This paper provides an overview of smart city (SC) standards, by describing the different international initiatives and how the SC standards support SC infrastructure, among other factors. This seems important as SC standards are an emerging topic without much discussion apart from indicators in the literature. However, this overview remains very superficial: it covers the basic details without ever going into the implications or applications of the standards. While a list of comments are provided below, two points are highlighted as essential to consider: 1) it is necessary to provide insight into why the smart city matters or the importance of the smart city (especially using references from the literature), and 2) it is highly recommended to provide one case study that encompasses the different components described in the paper or, after each component (e.g., Electric power infrastructure monitoring or Sensing and Data Collection System for City Infrastructure) several examples to position the content in the paper to real world cases. Without these two points, the research in the paper does adequately not connect with the smart city literature.

1.     The introduction states several facts and a gap in the literature but does not have any references. These statements require support from the existing literature.

2.     The introduction provides the context (need for smart city standards and idea that this paper will focus on the Chinese case) but does not provide a research question or objective. It is advised to write out the question or objective in the introduction.

Moreover, there is no transition between the first few general sentences about SC standards and the Chinese case. Why is China is being chosen as the empirical setting for the paper?

(EDIT: we later find out that it is not chosen as the empirical section, yet it presents like it will be.)

It would also be beneficial if the authors could add details and relevance concerning the smart city concept (why the SC? What is its importance? Has the literature already discussed standards in the SC? Etc.) It is telling that there are no SC papers in the references list.

Overall, I recommend the Introduction is re-written to better explain the rationale behind the paper and objectives of the paper.

3.     In Section 2.1., there is a presentation of which major International Standardization Organization published what and when, but there is no indication of what these different organizations are doing about SC standardization. It is recommended to briefly explain each initiative to offer a better comparison across the organizations.

4.     In Section 2.2., the focus shifts to the ITU FG SSC platform… but there is no indication as to why this particular one is selected for further analysis. This should be explained and justified in the paper.

5.     In Section 2.3., the focus is rather on “ITU-T”. The first time this term appears is on page 3, and prior to this there is no introduction. The authors can explain what the “T” stands for in ITU-T (just as they explained the abbreviation ITU at the beginning of the paper).

6.     China is used as an example in the Introduction section about the relevance of SCP… and therefore the reader expects a case study or similar about the implementation of standards in China. There is no such application in the paper. It is recommended that the authors provide a real-life example of a smart city platform. Otherwise, the paper remains very conceptual and unconnected to real cases.

7.     General comment: There is a heavy use of semi-colons (;) that makes the paper at times difficult to read (e.g., section 1 and 2.1). It is advised that the authors re-consider these sentences, with the recommendation they simplify the sentences by making them short, complete sentences. To provide an example for Section 1, a suggestion is:

The rationale is as follows. Without standards, the objectives of smart city construction cannot be determined. If there is no unified standard, it may not avoid repeated construction and waste of resources. Moreover, due to the lack of standards, it is difficult to evaluate the achievements of smart city construction.

I wish the authors good luck.

Reviewer 2 Report

Comments and Suggestions for Authors

The development of standards is an evolving process that can change over time. What is presented in the article, is it materialized, in part, by the publication of some standards. Does the preliminary stage of developing standards allow to apply it? Can some aspects be added regarding this? If some real example of standardization of urban infrastructure can be added will give more pragmatic aspects of paper. Another problem is how about happend if standards are not followed? What the consequences may be? These standards may be imposed? What are the most advanced countries in applying and developing these standards today?

Reviewer 3 Report

Comments and Suggestions for Authors

The paper titled Progress of Standardization of Urban Infrastructure in Smart 2
City.
presents a good topic, but not well written, and should be improved. Below some comments that need to be addressed.

-        The abstract needs to be reworked : There is no comment about the novelty of this work. What is the main contribution of this work.

-        Introduction:  There is an important part of any study of smart city with its concept definition and evolution. This aspect should be added as a first paragraph in your introduction part.

-        In overall introdution should be improved, it contains  obvious statements without any reference!!!.

-        The current version contains a lot of abbreviations.I suggest to add a table for these abbreviations and delete all the abbreviations used in the text of your manuscript.

-        There is a lack for published papers dealing with the same concept in this paper. More intention should be given to litterature review.

-        Lines 229-230 , add a reference.

-        Line 26 (; moreover,) should be. Moreover,

-        English should be improved

The manuscript still requires a revision. Major revision is required.

Good luck

Round 2

Reviewer 1 Report

Comments and Suggestions for Authors

The authors have answered or responded to all my previous concerns. After the revisions, I believe the paper is ready for publication.

Reviewer 3 Report

Comments and Suggestions for Authors

The paper  has been improved, the authors have added the information requested in the first revision. I recommend the paper for publication
